

# Landscape-scale water balance monitoring with an iGrav superconducting gravimeter in a field enclosure

Andreas Güntner[1,4], Marvin Reich[1], Michal Mikolaj[1], Benjamin Creutzfeldt[2], Stephan Schroeder[1], Hartmut Wziontek[3]

[1]Helmholtz Centre Potsdam GFZ German Research Centre for Geosciences, Section 5.4 Hydrology, 14473 Potsdam, Germany
[2]Senate Department for Urban Development and the Environment, 10707 Berlin, Germany
[3]Federal Agency for Cartography and Geodesy (BKG), 04105 Leipzig, Germany
[4]University of Potsdam, Institute of Earth and Environmental Science, 14476 Potsdam, Germany

*Correspondence to*: Andreas Güntner (guentner@gfz-potsdam.de)

**Abstract.** In spite of the fundamental role of the landscape water balance for the Earth's water and energy cycles, monitoring the water balance and its components beyond the point scale is notoriously difficult due to the multitude of flow and storage processes and their spatial heterogeneity. Here, we present the first field deployment of an iGrav superconducting gravimeter (SG) in a minimized enclosure for long-term integrative monitoring of water storage changes. Results of the field SG on a grassland site under wet-temperate climate conditions were compared to data provided by a nearby SG located in the controlled environment of an observatory building. The field system proves to provide gravity time series that are similarly precise as those of the observatory SG. At the same time, the field SG is more sensitive to hydrological variations than the observatory SG. We demonstrate that the gravity variations observed by the field setup are almost independent of the depth below the terrain surface where water storage changes occur (contrary to SGs in buildings), and thus the field SG system directly observes the total water storage change, i.e., the water balance, in its surroundings in an integrative way. We provide a framework to single out the water balance components actual evapotranspiration and lateral subsurface discharge from the gravity time series on annual to daily time scales. With about 99% and 85% of the gravity signal due to local water storage changes originating within a radius of 4000 and 200 meter around the instrument, respectively, this setup paves the road towards gravimetry as a continuous hydrological field monitoring technique at the landscape scale.

## 1 Introduction

Water storage is the fundamental state variable of the global water cycle. It is a key state that governs processes of land-atmosphere water and energy exchange, runoff generation, groundwater recharge, as well as matter and solute transport in the Earth's biogeochemical cycles. Quantifying water storage is the basis for water resources assessment and management. Water storage dynamics reflect the net effect of all water fluxes acting in the landscape, balancing precipitation, evapotranspiration and runoff. It has been suggested for a long time that direct measurements of total water storage variations are needed for



closing the water budget at spatial scales of practical relevance such as the forest stand, landscape or catchment scale, and for understanding the relationships between storage and water fluxes (Beven, 2002; Davies and Beven, 2015).

The major obstacles for integrative monitoring of water storage variations at the field or landscape scale are, first, that total water storage is a complex state of the hydrological system, composed of various individual storage compartments that would need to be monitored individually. This includes interception storage, soil moisture, vadose zone, groundwater, surface water bodies, snow and ice, with varying contributions depending on the environmental and climatic conditions (e.g., Güntner et al., 2007). Secondly, considerable heterogeneity even at small spatial scales makes it challenging to infer representative storage dynamics at larger scales from traditional point-scale measurements. While progress has been made during the last years with satellite-based and geophysical methods at larger scales (Ochsner et al., 2013; Bogena et al., 2015), these techniques measure the soil moisture component with limited integration depth only. Total water storage variations are available from satellite gravimetry at regional to continental scales (Tapley et al., 2004), however, with low spatial and temporal resolution. Terrestrial gravimetry, in turn, i.e., measuring with gravimeters on the ground (see Crossley et al. (2013) and Niebauer (2015) for an overview), is an emerging technology for non-invasive monitoring of water storage variations at the landscape scale in an integrative way over all storage compartments (Bogena et al., 2015).

Terrestrial gravimetry is the measurement of the acceleration of gravity at the Earth's surface, varying in space and time according to Newton's law of mass attraction and due to the Earth's rotation. Relative gravity changes are determined by measuring the impact of the resulting force changes on a test mass. In absolute gravity measurements, the magnitude of the gravity vector is deduced by observing the trajectory of a free moving object along the vertical. Relative gravity measurements can be carried out continuously, while current technology in absolute gravimetry is restricted to periodically repeated observations (time-lapse measurements). For the continuous monitoring of the Newtonian mass attraction due to water mass changes in the surroundings of the instrument, which is about seven orders of magnitude smaller than the attraction by the Earth mass, most stable and sensitive relative gravimeters are required. Even though todays spring type gravimeters are well advanced, superconducting gravimeters (SGs) show highest sensitivity and long term stability (Neumeyer, 2010; Hinderer et al., 2015). In SGs, the conventional spring-mass system is replaced by a superconducting sphere that floats in the magnetic field generated by superconducting coils.

Time-lapse gravity measurements have been applied in hydrology, for instance, for studying karst systems (e.g., Jacob et al., 2009), analysis of water flow and storage processes at the hillslope and small catchment scale (e.g., Creutzfeldt et al., 2010a; Hector et al., 2013; Pfeffer et al., 2013; Piccolroaz et al., 2015), or groundwater model calibration (e.g., Christiansen et al., 2011). Despite recent improvements of processing strategies towards hydrological applications (Kennedy and Ferre, 2016), the use of time-lapse gravimetry is limited by instrument accuracy and low temporal resolution. High-precision and time continuous monitoring of gravity variations with SGs has been shown to be sensitive enough to resolve water storage variations at seasonal and event time scales occurring within a radius of few 100 meters around the instrument (Creutzfeldt et al., 2008). To this end, all non-hydrological gravity effects, in particular tidal variations, atmospheric changes and polar motion have to





be carefully removed (Hinderer et al., 2015), as these signals are up to two orders of magnitude larger than the signal of interest. The same applies to seasonal large scale hydrological effects, both in terms of their mass attraction effect as well as in terms of continental loading variations, i.e., deformation of the Earth surface (Boy and Hinderer, 2006). Continuous measurements with SGs have been used in hydrology to study local water storage variations (e.g., Creutzfeldt et al., 2010a; Hector et al.,

2014; Fores et al., 2017), validation and calibration of hydrological models (e.g., Naujoks et al., 2010), or for unraveling the long-term effects of hydrological extremes (Creutzfeldt et al., 2012). Recently, Van Camp et al., 2016a presented the possibility of monitoring evapotranspiration with SGs, albeit limited in their study by the need to stack the gravity time series over several years in order to isolate the signal, an unusual and for most other cases impractical deployment of the SG in a cave about 50m below the terrain surface, and by the difficulty of correcting for mass changes due to lateral subsurface flow processes.

In spite of these and other studies, the potential of superconducting gravimetry for hydrological applications has not yet been fully explored. Main reasons are that, first, the SG monitoring sites have rarely been selected or optimized for the purpose of monitoring water storage, but for other geodetic or geophysical interests. Secondly, given the highly sensitive SG technology and the instrument size, SGs usually are permanently installed in buildings or in underground observatories under temperature controlled and low-noise conditions. Thus, a field deployment of a SG as a hydrological sensor has been beyond of what was

feasible from a technological point of view. And third, the hydrological interpretation of the SG time series is in most cases hindered by disturbance of natural local hydrology due to the observatory building itself and the distance of the hydrological variations of interest to the instrument. While the last aspect known as umbrella effect (Creutzfeldt et al., 2010b; Deville et al., 2013) is particularly relevant because of the important effect of mass changes in the near-field (several meters) around the gravimeter, it is usually ignored in view of missing moisture measurements and unknown flow dynamics below or above the

observation room. In the present study, we minimize the umbrella effect by deploying the SG in a compact field enclosure. The latest generation of superconducting gravimeters (iGrav by GWR Instruments, Inc.; Warburton et al., 2010) being smaller and lighter than previous observatory SGs facilitates this development. A prototype enclosure was used by Kennedy et al. (2014) in an arid environment. Here, for the first time, we test the performance and assess the hydrological value of a long-term SG field installation for the example of a grassland site under wet-temperate climate conditions.

## 2 Study site and hydrological data

The Geodetic Observatory Wettzell (Schlüter et al., 2007), operated by the Federal Agency for Cartography and Geodesy (BKG), is located on a mountain ridge in the Bavarian Forest of south-eastern Germany (Figure 1). The crystalline basement of metamorphic rocks (Gneiss) in Wettzell is covered from bottom to top by weathering zones of fractured Gneiss, saprolite, periglacial weathering layers, and soil, with Cambisols making up the predominant soil type. The climate of the study area is

temperate with mean annual precipitation of 995 mm and mean annual temperature of 7°C. Land cover in the surroundings of the observatory is dominated by a mosaic of grassland and forest, while grassland, gravel and sealed surfaces of roads and buildings alternate on the grounds of the observatory. For a detailed description of the environmental and hydro-meteorological



conditions of the study area, its hydrological dynamics including water storage variations and the hydro-meteorological monitoring systems including weather stations, clusters of soil moisture probes, groundwater observation wells, a lysimeter, and snow monitoring, see Creutzfeldt et al. (2010a) and Creutzfeldt et al. (2012).

**Figure 1: The Geodetic Observatory Wettzell, including the position of hydrological and gravimetric monitoring systems used in this study. Insert figure shows the region surrounding the observatory with Digital Elevation Model (DEM) with minimum (black) and maximum (white) elevation of 379 and 911 m, respectively**





At the Wettzell station, BKG operates two superconducting gravimeters of the observatory type (SG029 and SG030) in dedicated gravimeter buildings. At a distance of 41 m from SG030 (Figure 1), the new field setup with an iGrav has been installed in February 2015. The closest groundwater observation well (called BK3) is at a distance of 19 m from the iGrav

location as part of a network of 9 observation wells on the observatory area with continuous hourly water level monitoring. During the study period, the groundwater level at BK3 is 6.2 m below the terrain surface in average, with peak-to-peak amplitude of 3.5 m.

Time series of precipitation, air temperature, humidity, wind speed and net radiation are available from meteorological stations on the grounds of the observatory with hourly temporal resolution. Precipitation data obtained with a Hellmann-type gauge

have been corrected for systematic under-catch errors due to wind and evaporation effects by applying the approach of Richter, 1995 as recommended by the German Weather Service (DWD). The correction relies on two parameters (Table 1), one site-specific parameter accounting for wind exposition and an empirical exponent. Both parameters depend on the precipitation type. Precipitation is assumed to be snow at air temperatures below 0 °C, mixed precipitation between 0 and 4 °C, and liquid precipitation above 4 °C. The correction method is designed for daily precipitation. To correct the hourly values, each daily

correction volume is uniformly distributed over all hours with precipitation of the same day.

Since 2007, a weighing UMS lysimeter with a 1.5 m deep undisturbed soil monolith and a surface area of 1 m$^2$ with grass cover similar to the surrounding grassland sites is operated at the Geodetic Observatory (von Unold and Fank, 2008; Creutzfeldt et al., 2010b). For this study, the recorded time series of both the monolith weight and the drainage tank weight were filtered for noisy data. The correction method applied follows Peters et al., 2014 as part of their adaptive window and adaptive threshold

(AVAT) filter. It consists of a moving average smoothing routine where the window length is adjusted dynamically for each timestep. For the recorded lysimeter time series with a temporal resolution of 1 minute, we used values of 1 to 7 for the possible order of the fitted polynom, and a maximum window width of 31 minutes. The filtered lysimeter data were aggregated to one hour resolution, from which time series of precipitation and actual evapotranspiration were extracted by considering the overall lysimeter system weight (sum of both monolith and drainage tank) at each point in time, defining precipitation and

evapotranspiration as an increase or decrease in weight, respectively. Furthermore, independent from the lysimeter, grass reference evapotranspiration was calculated from the available meteorological data with the Penman-Monteith approach following the FAO-56 standard (Allen et al., 1998).

Streamflow time series were available at the gauging station Chamerau of river Regen, the main river that drains the mountainous area of the Bavarian forest surrounding the observatory. Wettzell is located in a headwater catchment contributing

to the river Regen, at a distance of about 10 km to the gauging station. The total catchment area at the Chamerau station is 1356 km², mean streamflow is 26.2 m³ s$^{-1}$ (period 1931-2013) (Bayerisches Landesamt für Umwelt, 2016)



## 3 Gravimeter system and performance

### 3.1 Configuration of the gravimeter (iGrav) field monitoring system

The monitoring system is a two enclosure system, comprising the SG itself in its dome-shaped field enclosure on the one hand (iGFE – iGrav field enclosure), and an external box for peripheral hardware on the other hand (iBox) (Figure 2). The instrument
with serial number #006 (iGrav006) was deployed here. A SG records time series of gravity variations as voltage changes in an electronic feedback loop. This feedback loop keeps the levitated superconducting sphere in a constant position by adjusting an ultra-stable and homogenous magnetic field to compensate external force changes. The magnetic field is generated by Niobium coils. The whole sensor system is temperature stabilized by liquid Helium at about 4.7 Kelvin. The system is actively cooled by cryocooler which is re-liquefying evaporating Helium gas and enables a closed system with only 16 litres of liquid
Helium.

The main function of the iGFE field enclosure is to protect the iGrav from environmental effects, in particular humidity in form of precipitation and dew, and wind load, and to provide a stable and isolated casing for efficient temperature control in its interior. The enclosure with an outer diameter of 0.9 m is made of double aluminium walls with isolation foam in between. The iGrav (baseplate and body diameter of 0.55 and 0.36 m, respectively) and the field enclosure were mounted separately on
a concrete pillar in such a way that there is no transfer of enclosure vibrations and deformations, due to wind stress for instance, to the instrument. Similarly, to minimize noise transfer to the sensor unit, the cryocooler was attached to the field enclosure (via the red platform in Figure 2a). The pillar itself has a cylindrical shape with a total height of 2 m, thereof 0.8 m above the ground. The diameter of the pillar is 1 m, leaving only little space around the enclosure. For maintenance, the instrument can be accessed from top via a removable cap (Figure 2a). The iGFE also houses controllers with temperature sensors ranging
from the bottom to the top, as well as a PC, heating and cooling grills. The cooling grills are connected to a water chiller inside the iBox. The iBox also contains the compressor to drive the cryocooler, a gas bottle for re-liquefying Helium in the iGrav or recharging the compressor, the power supply including an UPS backup system, controller, temperature sensors and a PC for the remote control of the monitoring system (Figure 2b). The temperature inside the iBox is passively regulated with fans. The umbrella effect of the iBox on the gravity observations is negligible, as the footprint is less than 2 m² and the length of
connection hoses for cooling water and Helium gas allows a distance of up to 15 m from the gravimeter.



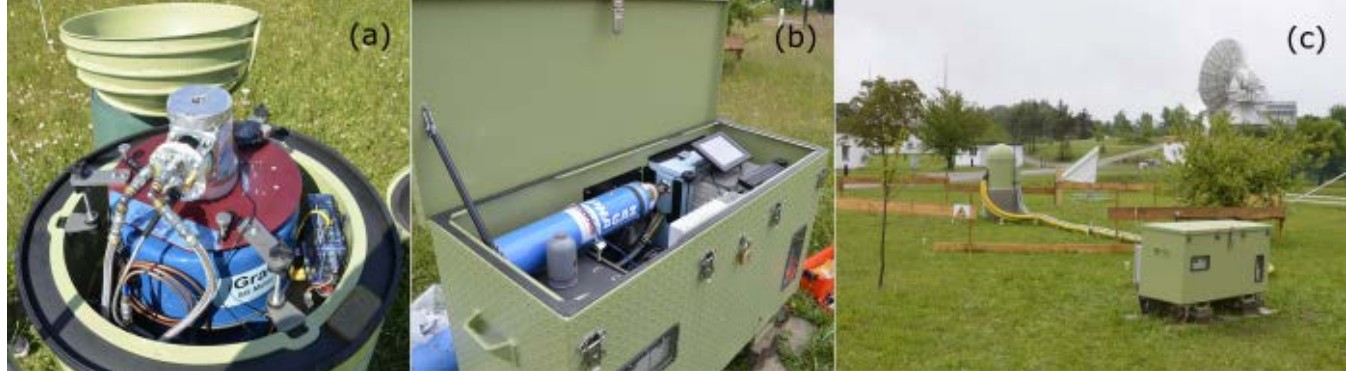

**Figure 2: Photographs of the gravimeter monitoring system with (a) the iGrav inside the iGFE field enclosure, (b) peripheral hardware inside the iBox, and (c) an overall view of iBox (foreground), dome-shaped iGFE, and the yellow connection hoses. Photos taken on a grassland site of the Geodetic Observatory in Wettzell, Germany.**

### 3.2 iGrav data processing

The voltage changes measured by the gravimeter have to be transformed into a gravity signal by calibration. Usually, the respective scale factor of the SG is determined by co-located measurements of a dominant gravity signal at daily time scales, i.e., the tidal variations with either an absolute or a well calibrated relative gravimeter (e.g., Meurers, 2012; Van Camp et al.,

2016b). Although two well-calibrated SG are operated nearby, it was decided for the present study to determine the scale factor and phase delay by regression and cross-correlation against a tidal model resulting from the harmonic analysis of a nine years gravity record by SG029 at the same site. This pragmatic approach effectively minimized any tidal residuals in the gravity time series after reducing the tidal signal, in particular at diurnal to semi-diurnal frequencies which may interfere with hydrological mass variations at these frequencies, especially evapotranspiration. To validate this approach, the same procedure

was applied to the nearby SG030. The scale factor computed in this way differed by only 0.8 per mill from the value obtained from calibration using absolute gravity measurements. The calibration factor for iGrav006 used here was 914.416 ±0.005 nm s$^{-2}$ V$^{-1}$ with a phase delay of 11.7 seconds.

As the hydrological signal rarely exceeds 10% of the total measured gravity signal, other gravity effects caused by Earth and ocean tides, Earth rotation and atmospheric variations have to be carefully removed. The local tide model mentioned above

was used and atmospheric effects were corrected for with Atmacs (Atmospheric attraction computation service, Klugel and Wziontek, 2009), supplemented by in-situ observations of atmospheric pressure variations to enhance the temporal resolution. The polar motion effect was computed based on the Earth orientation parameters provided by IERS (International Earth Rotation and Reference Systems Service, www.iers.org). For further analysis, the 1 second gravity time series was filtered using a low-pass filter and decimated to 1 minute temporal resolution. Nine steps found in the gravity residuals due to unknown

spurious instrumental effects were corrected manually via visual inspection. This includes two critical steps (in June 2015 and May 2016) that occurred during rain events so that the instrumental error could not be separated unequivocally from a




hydrological mass effect. Hence, a small uncertainty with respect to the level of the residual time series after these events remains.

The instrumental drift of a SG is usually obtained by repeated co-located absolute gravity observations over longer time spans, assuming identical gravity variations at both sensor locations. At the iGrav site, measurements with an absolute gravimeter

could not be carried precisely enough under the field conditions. Drift determination for iGrav006 based on the drift corrected signal of SG030 was not possible either, as hydrological near-field effects turned out to be too different at both locations. Therefore, the drift of iGrav006 was estimated based on two epochs for which the same total water storage was estimated from independent observations. We assume here that total water storage is the sum of (i) soil moisture storage observed by the lysimeter and (ii) groundwater storage derived from the groundwater level observed at BK3. Based on these data, the same

total water storage was found for the days 19 May 2015 and 12 April 2016. This resulted in an iGrav006 drift rate of +94 nm $s^{-2}$ $yr^{-1}$. This drift rate is higher than a first long-term drift of 45 nm $s^{-2}$ $yr^{-1}$ derived for iGrav002 from precise absolute gravity measurements over a four year period by Fores et al. (2017). The difference is not surprising as it is known from observatory SGs that drift rates vary among the individual instruments (e.g., Crossley et al., 2013). However, an uncertainty of the drift estimation in our study originates from neglecting possible storage changes in the vadose zone in between the lower boundary

of the lysimeter and the groundwater level. The drift was removed from the gravity time series before further analysis. Furthermore, all iGrav006 measurements recorded before 01 May 2015 were discarded because of vibration effects due to inadequate initial mounting of the cryocooler, and several steps related to system maintenance during the initial phase of the field deployment. Figure 3 shows the final time series of the gravity residuals of iGrav006 in comparison to the observatory SG030. These time series represent mainly hydrological mass effects as all other gravity effects have been removed as

described above.

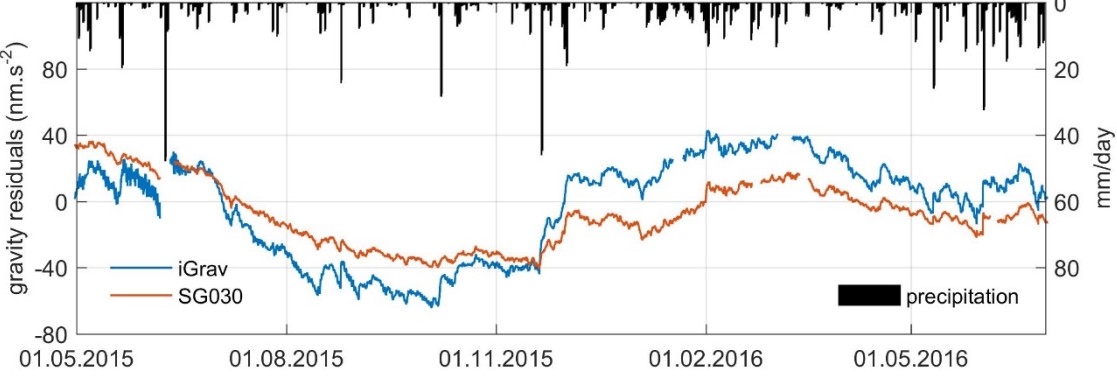

**Figure 3: Time series of gravity residuals of iGrav006 in the field enclosure and of SG030 in the observatory building, and daily precipitation rates (from top and right axis) at the Observatory Wettzell.**



### 3.3 System performance and data noise

One of the main technical challenges arising from the compact iGFE design under outdoor conditions is the efficient temperature control within the enclosure during all seasons of the wet temperate climate. After some minor modifications, the system was able to stabilize the temperature during most of the time.

The electronics board (PCB) is mounted below a sealed cover around the neck of the gravimeter and is flooded with Helium gas to avoid humidity. Stable temperature inside this casing is actively achieved by a heater for which a constant set point (here 32 °C) above the air temperature inside the iGFE was defined. Together with the general heating system of the iGFE, this heater showed sufficient performance to keep the PCB temperature constant during the winter season with minimum outside air temperature of -13°C. However, an unwanted temperature increase inside the iGFE was observed in particular

during warm summer days with high insolation. Under these conditions, the performance of the water-based cooling system with regard to the redistribution of cooled air within the iGFE was not sufficient. The resulting increase of the temperature on the PCB was found to have direct effects on the recorded gravity data. A PCB temperature increase caused an apparent decrease of gravity. Significant diurnal temperature variations of several degrees inside the iGFE exceeded the set point of the PCB temperature control and thus translated into PCB temperature patterns and related diurnal variations in the gravity time series.

To determine a regression parameter between PCB temperature and gravity, the PCB temperature was artificially increased via the control software. This experiment showed a non-linear response, i.e., the regression parameters varied between -4.4 and -3.2 nm/s$^2$ per °C temperature increase for different temperature levels. Thus, a direct correction for the spurious diurnal temperature effects was possible with low accuracy only. In July 2016, the PCB temperature issue could be solved by installing extra fans inside the field enclosure and by increasing the set point of the PCB temperature regulation to 34 °C. The fans ensure

a better circulation of the cooled air inside the iGFE and thus avoid PCB temperatures exceeding the set point even during hot summer days (Figure 4). In turn, no disturbing effect on gravimeter noise due to the fans could be observed. It should also be noted that the temperature variations inside the field enclosure causes tilt effects due to thermal expansion of the mount of the gravimeter and heat transfer from the thermal levellers. This resulted in notable variations in the control values of the active tilt feedback system which was able to compensate these effects.

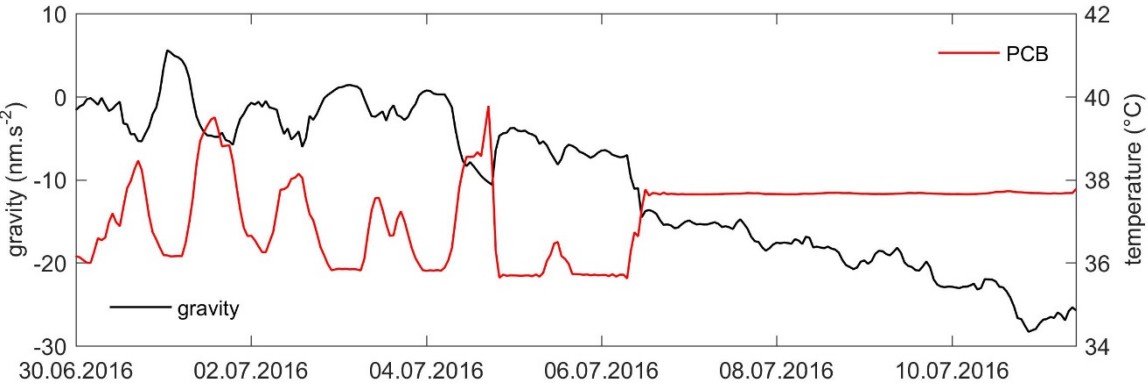





**Figure 4: Comparison of gravity residuals and PCB temperature (electronics board) before and after improvement of temperature control inside the field enclosure on July 7, 2016. Note that there is an offset of about 4 °C between the PCB set point temperature (34°C) and the actually recorded temperature.**

To characterize the performance of iGrav006 in the field enclosure, its noise level was compared with those of the nearby dual sphere gravimeters SG029 and SG030, both located in a controlled environment of buildings at the observatory. As a common quality indicator for the sensor, the power spectral density (PSD) of the gravity time series is considered at periods from 3 hours to 1 minute in the (sub-)seismic frequency range (e.g., Banka and Crossley, 1999, Rosat and Hinderer, 2011). While the noise is the combination of instrument and site noise, at frequencies larger than about 1 mHz the instrument noise tends to be higher

than seismic or environmental noise. Similar to the procedure described in Rosat and Hinderer, 2011, tidal and atmospheric effects were removed by the described models before the PSD was estimated by averaging 12 segments overlapped by 75% from a period of 6 quiet days with low seismic activity (Feb 24th to 29th 2016). The results for all five sensors are shown in Figure 5, together with the New Low Noise Model (NLNM) of Peterson, 1993 as a reference for the lowest background noise levels of the Global Seismographic Network. The noise level of iGrav006 is very similar to those of the observatory gravity

for most parts of the spectrum, with slightly higher values at periods longer than half an hour. This demonstrates the high performance of the iGrav sensor and the quality of the field enclosure system with reasonable reduction of cold head vibrations and no visible additional noise from environmental effects such as wind load. The slightly higher PSD values at hourly scales may indicate very small diurnal temperature effects or a higher sensitivity of the field system to hydrological variations. The peaks at periods shorter than a few minutes indicate the resonance frequency (parasitic modes) of the respective sensor, which

were not excited for all spheres during the analysis period.



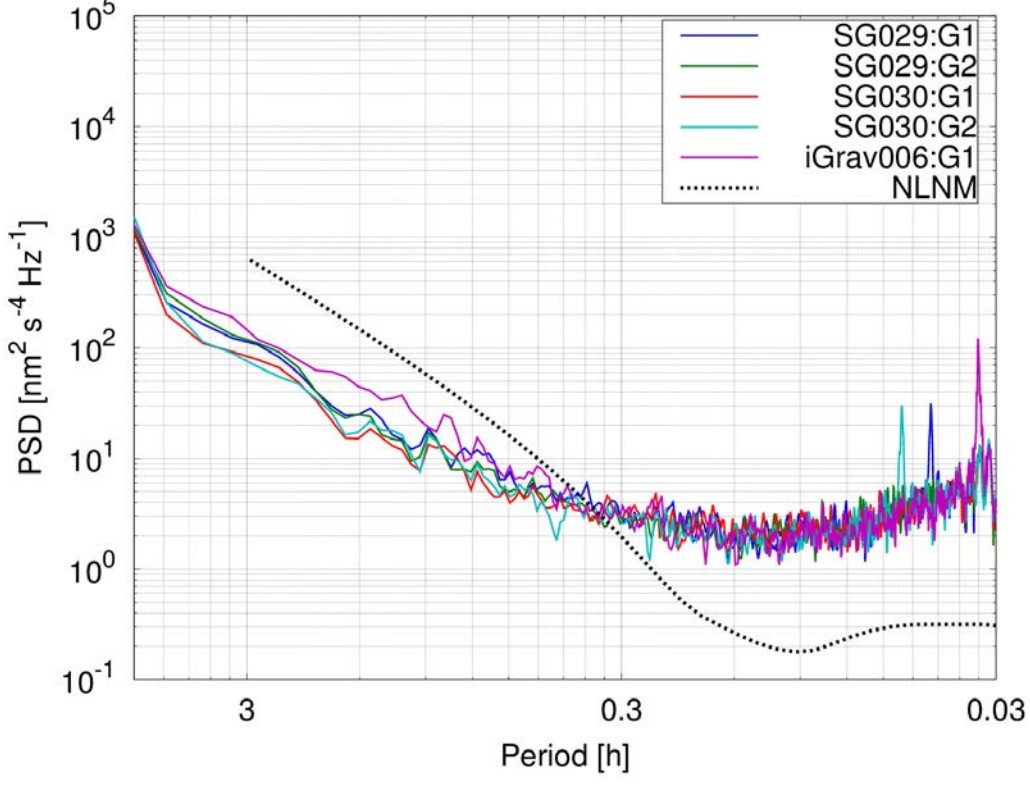

**Figure 5: Noise level of the dual sphere observatory gravimeters (SG029 and 030) with sensors G1 and G2, and of gravimeter inside the field enclosure (iGrav006), expressed as the power spectral density (PSD) for the time period 24 February 2016 to 29 February 2016. The New Low Noise Model (NLNM) has been added as a reference for minimum seismic noise.**

## 4 Hydrological value

### 4.1 Sensitivity to water storage variations

It is well known that the gravitational force reduces by the square of the distance from the source. Furthermore, a gravimeter is sensitive only in the direction of the vertical, measuring the magnitude of the gravity vector and is therefore insensitive to

10    forces due to mass changes perpendicular to it. Both aspects are important if the sensitivity of a gravimeter to water storage variations is considered. Following Bonatz, 1967, a first approximation can be given by neglecting topography and assuming a homogenous layer where water storage changes occur. The mass attraction effect $g_c$ can then be described by a homogenous cylinder with the sensor on its symmetry axis following Eq. (1) (Heiskanen and Moritz, 1967):

$$g_c = 2\,\pi\,G\,\rho\left[d + \sqrt{r^2 + h^2} - \sqrt{r^2 + (h+d)^2}\right], \tag{1}$$

15    where $d$ is the thickness and $r$ the radius of the cylindrical layer. $h$ is the distance of the sensor along the symmetry axis (i.e., the height of the sensor above a soil layer where water storage changes occur), while $\rho$ is the density of the layer (as a function





of its water content), and $G$ the universal gravimetric constant. Increasing the radius to infinity results in the well-known Bouguer plate following Eq. (2):

$$g_c = 2 \pi G \rho d \,, \tag{2}$$

for which the mass attraction effect only depends on the thickness of the layer and on its density. Accordingly, if the radius of

the region is chosen sufficiently large, the gravity effect does not depend on the distance of the sensor to the layer. The solid lines in Figure 6 illustrate that this is the case for a radius of about 100 m to 200 m for sensor heights of up to 5 m above the cylinder, as the resulting gravity effect converges asymptotically to the effect of the Bouguer plate (dotted purple line). These results changed significantly if the concrete monument (gravimeter pillar) on which the gravimeter is installed were considered. Assuming no water storage changes within the pillar volume, the total gravity effect reduces considerably for sensor heights

below 1 m and the effect of an infinite plate is never reached. However, for sensor heights above 2 m both curves come very close at the same radius, since the effect of the monument decreases rapidly with distance.

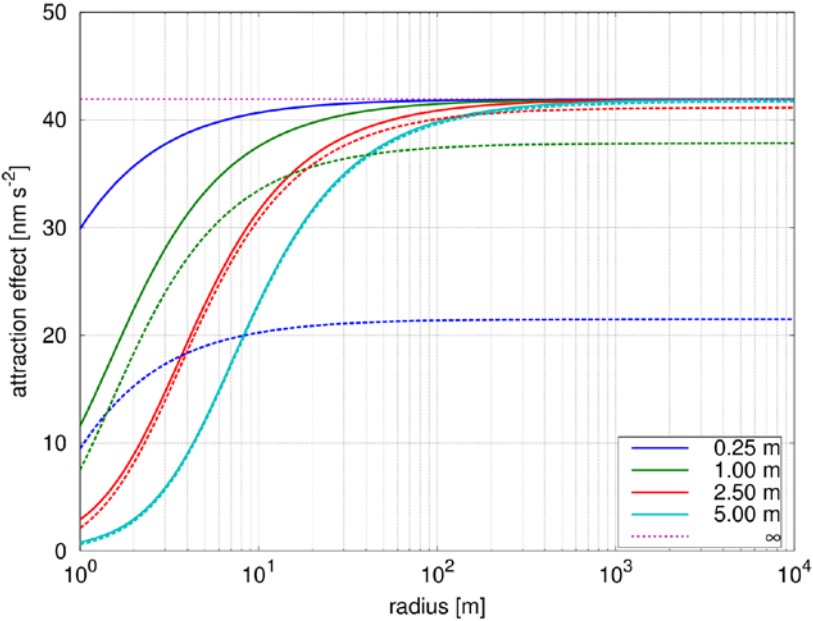

**Figure 6: Gravity effect of a homogenous cylinder with a thickness of 10 cm, a density of 1000 kg m$^{-3}$ and varying radii on a sensor**
**placed at different heights (0.25 m to 5 m) above the cylinder (solid lines). Dashed lines show the same gravity effects but reduced for a cylinder with a radius of 0.5 m and a depth of 1.2 m, corresponding to the dimensions of the pillar used for the installation of iGrav006 at Wettzell.**

Thus, amplifying the gravity signal that is recorded by a gravimeter due to water storage variations in its surroundings can basically be achieved in two ways: (i) by reducing the sealed area of pillar and housing around the gravimeter (i.e., minimizing

the umbrella effect), and (ii) by positioning the gravimeter sensor in a suitable position within the local topography. While (i) is the main motivation for the compact design of the field enclosure system described in this study, (ii) has also been considered with the iGrav deployment at the Wettzell site. Both issues are discussed in the following in comparison to the nearby



observatory SG030. For the calculation of the gravity effect on the gravimeter sensor, a gravity model with a prism approximation was used (Nagy, 1966). The location of each prism with respect to the gravimeter sensor is defined by a high-resolution local digital elevation model. The size of individual prisms is smaller the closer they are to the sensor. Given the location and the water mass change in the prism, the gravity effect of each prism on the sensor can be integrated analytically

based on Newton's law of mass attraction, and finally summed up for all prisms to get the overall gravity effect of water storage changes in the surroundings of the gravimeter.

The area sealed by foundations and the roof of the observatory building of SG030 equals 88 $m^2$ while the iGrav pillar covers about 0.8 $m^2$. Soil moisture sensors installed beneath the SG030 building show that soil moisture variations in the first two meters below the building are absent or markedly smaller than for outside sensors under natural conditions in the soil

surrounding the building (Reich et al., in preparation). As an example, a water storage change of 10 mm in the first two meters below the SG building and below the iGrav pillar would cause a gravity effect of 2.79 nm $s^{-2}$ and 0.15 nm $s^{-2}$ for SG030 and iGrav006, respectively. In other words, a natural storage change of 10 mm will result in a gravity signal that is about 18 times smaller for SG030 than for iGrav006 due to the umbrella effect of housing or pillar. For the further analysis in this study, we set the depth of the umbrella space, i.e. the depth below housing or pillar in which no soil moisture variations take place, to 2

m.

The topographic effect reflects the spatial distribution of hydrological mass changes outside of the building or pillar relative to the position of the gravimeter sensor. While the building with SG030 is located in a topographically low position, the iGrav was intentionally placed on an adjacent upslope location. For SG030, water storage changes partly take place at topographic positions above the gravimeter sensor with a gravity effect that is opposite in sign to the same changes occurring

topographically below the gravimeter sensor. In total, the gravity effects of near-surface soil moisture variations in the landscape cancel out to some extent for SG030. In contrary, for iGrav006 all near-field mass changes are located below the gravity sensor so that no cancelling effect occurs. Furthermore, following the theoretical considerations above (Figure 6), the instrument was placed on a pillar of 0.8 m height, leading to an effective height of the gravity sensor of 1.05 m above the terrain surface. This further amplifies the sensitivity of the instrument to near-surface soil moisture variations (Figure 7). While

this sensitivity increases markedly within the first meter, it levels off at even higher sensor positions. Similarly, the increase of sensitivity with sensor height is less pronounced if the water storage variations occur in larger soil depths (Hector et al., 2014). In this study, the chosen sensor height is a compromise between signal sensitivity, on the one hand, and system operability with regard to ease of access to the field enclosure and the gravimeter, and stability of the concrete pillar, on the other hand.






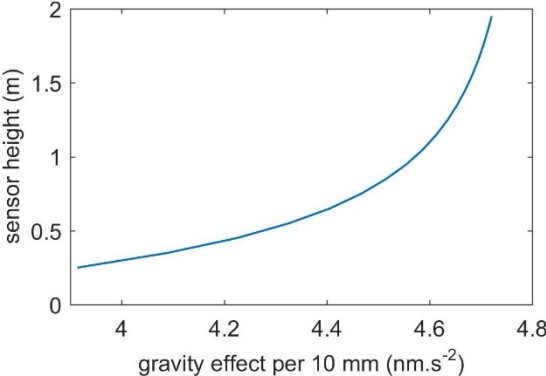

**Figure 7: Gravity effect of a water storage change of 10 mm in the uppermost soil layer (1 m thickness, uniformly distributed) as a function of the height of the gravity sensor above the terrain surface, for the iGrav location at Wettzell.**

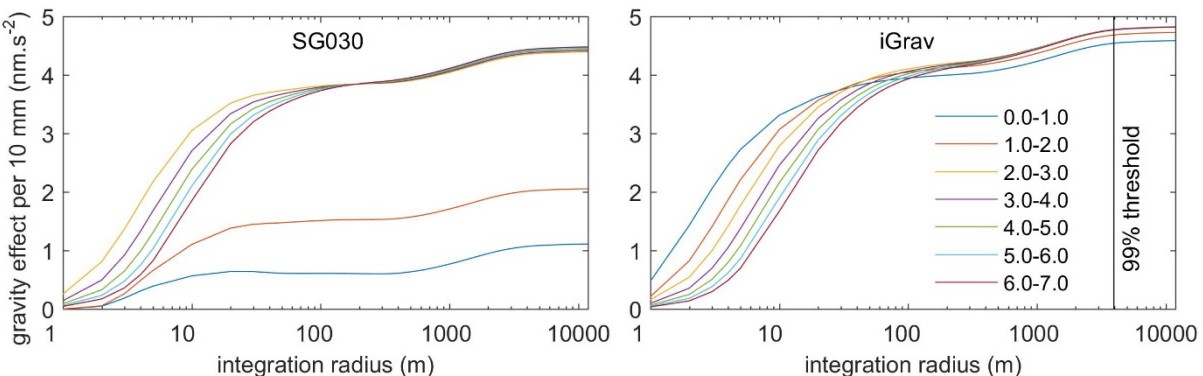

**Figure 8: Gravity effect of a 10 mm water storage change in different depths below the terrain surface (0 – 7 m depth) at Wettzell, considering the real topography and the umbrella effect of SG030 gravimeter building and the iGrav pillar, respectively (no storage change within 2 m underneath building and pillar).**

Using the prism-based gravity model, the gravity effects on SG030 and iGrav006 for water storage variations in different depths below the terrain surface are shown in Figure 8 for different integration radii, i.e., the distance from the gravimeter that is considered for calculating the gravity effect. All soil layers are assumed to be parallel to the surface, follow the topography given by the elevation model for the entire domain, and the water is uniformly distributed inside each layer. Both the umbrella effect and the topographic effect as explained above are considered for these calculations. As a result, soil moisture variations in the near-surface layers (0-2- m) have a considerably smaller gravity effect for SG030 than for the iGrav due to the umbrella effect. For soil moisture variations at larger depths, the iGrav also exhibits a slightly larger gravity effect than SG030 due to the topographic influence, but the difference between the two gravimeters is much smaller than for near-surface mass effects. Similar to Creutzfeldt et al. (2008), Figure 8 also shows that most of the local gravity effect originates from a distance of up to about 200 m around the instrument, which is consistent with the simple cylinder model described above. Increasing the





integration radius beyond 4 km has only a minor impact on either iGrav or SG030 because at this distance the gravity effect reaches 99% of the total effect computed for radius of 12 km. Here, only the mass attraction effect but not the surface deformation caused by large-scale hydrological loading of the Earth crust were considered. The overall higher sensitivity of the iGrav in the field enclosure to water storage changes is also expressed in markedly higher gravity amplitudes of iGrav006

in the residual gravity time series, both for event and for seasonal time scales (Figure 3).

The most interesting result of the sensitivity analysis shows up when comparing the gravity effects at the maximum integration radius (or in approximation at any radius at the landscape scale of larger than about 200 m that integrates over most of the gravimeter signal) (Figure 8). In this case, for iGrav006, the gravity effect of each layer is almost identical regardless of its depth. For example, the effect of a 10 mm water storage change is 4.6 and 4.8 nm s$^{-2}$ for the uppermost layer and the deepest

(groundwater) layer, respectively. For SG030, in contrast, the effect is 1.1 and 4.4 nm s$^{-2}$, respectively. This means that the iGrav006 in its field enclosure setup is rather insensitive to the depth below the terrain surface where the water storage change occurs, as the footprint of the monument is rather small and the sensors position sufficiently high. In turn, this means that once the water has infiltrated into the soil and increased the water storage, the vertical redistribution of water by hydrological flow processes does not influence the observed gravity signal, unless the water exits the domain again by evapotranspiration or by

lateral flow.

We confirm and illustrate this feature by a virtual experiment using a hydrological model, based on HYDRUS-1D (e.g., Simunek et al., 2016). The vertical extent of the model domain of 10 m is discretized into 1 cm intervals. A highly conductive sandy-loamy soil with a saturated hydraulic conductivity of 5.5 *10$^{-4}$ m s$^{-1}$ was chosen for the entire profile. The boundary conditions were set to "atmospheric" for the upper boundary and "no flow" for the lower one. The model was driven with an

artificial precipitation input over a period of 15 days and total sum of 361 mm of rain (24 mm d$^{-1}$). In model run 1, a constant evaporation rate of 12 mm d$^{-1}$ was set for the following 15 days. In model run 2 with the same precipitation for the first 15 days, zero evaporation was set for the following 15 days. No groundwater variations were considered in this experiment. The simulated profile soil moisture variations were then converted into gravity effects for the locations of both gravimeters SG030 and iGrav006 using the prism-based approach mentioned above. To this end, the simulated 1D soil moisture variations were

transferred to the entire domain and the real topography and building or pillar dimensions of Wettzell including an umbrella effect of 2 m in depth as described above were considered. The simulated profile soil moisture changes and related gravity effects are shown in Figure 9. The continuous wetting front advancement to larger depths during the entire experiment is obvious, as well as the drying topsoil layers due to evaporation in model run 1 after day 15.

The storage increase by precipitation and the subsequent decrease by evaporation cause a close to linear gravity

increase/decrease for iGrav006 in the field enclosure. The advancement of the wetting front to larger depths and the redistribution of water within the soil profile does not change the gravity signal for iGrav006. This can be clearly seen for model run 2 after day 15 where in the absence of evaporation or precipitation the total water storage in the system remains constant, as does gravity in the case of iGrav006. In contrast, the redistribution of water within the soil profile causes a further increase of gravity for SG030 even without net mass change, because the wetting front advancement moves water from top



soil layers with lower gravity sensitivity for SG030 due to the umbrella effect of the observatory building to deeper layers with higher sensitivity. As a consequence, the water mass loss due to evaporation after day 15 in model run 1 is not visible for SG030 as it is masked by the water redistribution in the profile that even causes an increase of gravity during the first days after evaporation kicked in. The complex interplay of (i) the hydrological processes of water redistribution within the profile

with (ii) the varying sensitivity to hydrological mass changes in depth due to the umbrella effect causes a non-linear gravity response of SG030 from which it is difficult to disentangle the underlying water storage variations. Similar behaviour was observed during rain events using iGrav002 installed inside a building in the Larzac plateau, France (Fores et al., 2017). In contrast, the iGrav006 setup in the field enclosure allows for monitoring the variations of total water storage within its sensitivity domain without the need to know the vertical distribution of hydrological mass changes. It is thus an unprecedented

means of assessing the landscape water balance in an integrative way as the net effect of all water inflows and outflows.

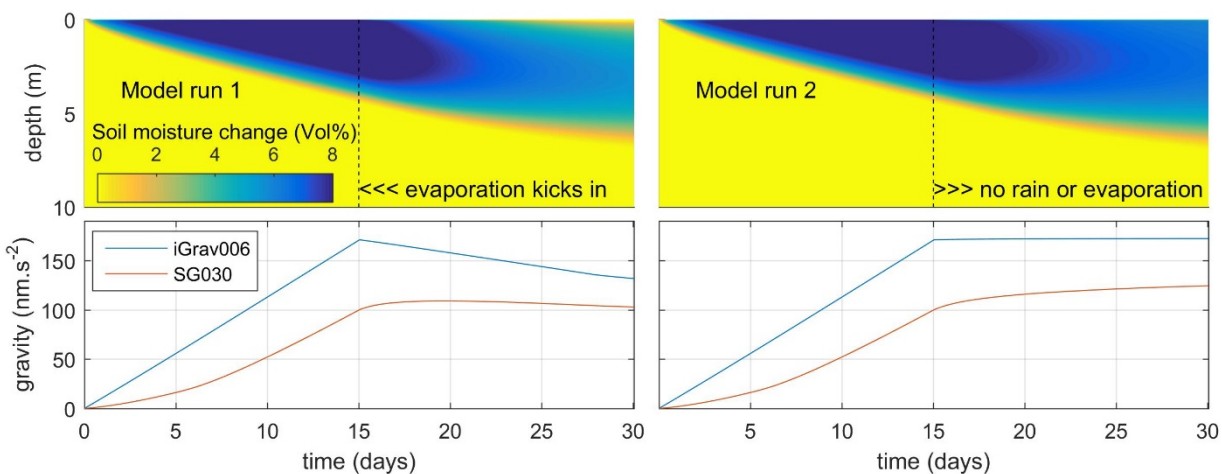

**Figure 9: Simulated profile soil moisture changes (upper plots) and gravity response for SG030 and iGrav006 at Wettzell (lower plots) for two model experiments, both with artificial rainfall during the first 15 days, and evaporation during the second 15 days**

**(model run 1 only).**

### 4.2 Resolving water balance components – annual scale

To demonstrate the value of the gravimeter field deployment for the direct analysis of the landscape water balance and its components, we set up the water balance equation in a way that the left-hand side of Eq. (3) is water storage change ($\mathrm{d}S/\mathrm{d}t$)

as given by the change of the iGrav006 gravity residuals ($\mathrm{d}g/\mathrm{d}t$). A constant mean sensitivity factor $s = 0.478$ nm s$^{-2}$ mm$^{-1}$ derived from the above sensitivity analysis (compare Figure 8) is used to convert a gravity change into an equivalent water storage change. It should be noted that the gravity residuals $\mathrm{d}g/\mathrm{d}t$ represent the gravimeter signal that was reduced for non-hydrological mass effects from Earth and ocean tides, Earth rotation and atmospheric variations. Thus, $\mathrm{d}g/\mathrm{d}t$ still comprises



gravity effects $(\mathrm{d}g_{glob}/\mathrm{d}t)$ from mass attraction and surface loading effects of non-local (i.e., continental to global scale) water storage variations which have to be removed for the present application. For this purpose, the mGlobe software package (Mikolaj et al., 2016) is used, considering simulated water storage variations on the global scale by four land surface models of the GLDAS system (Rodell et al., 2004).

The right-hand side of the water balance in Eq. (3) is composed of precipitation ($P$) minus actual evapotranspiration ($E$) minus runoff ($R$) from the area contributing to gravity variations seen by the iGrav006. As introduced in Sect. 2, as a first guess of the vertical fluxes at the land surface, $P$ was taken from local gauge measurements with under-catch correction. For $E$, the potential evapotranspiration without water limitation in the form of the Penman-Monteith grass reference evapotranspiration ($E_{ref}$) was taken because it can be quantified from meteorological observables alone. In view of the predominance of grassland

and partly forest with high infiltration capacity in the surroundings of the gravimeter location, and based on own field observations during rainfall events, surface runoff is considered to be negligible at the site so that $R$ encompasses subsurface runoff only. Given the specific topographic situation of the observatory on a ridge of the hilly mountain range, negligible lateral subsurface inflow is expected for the site because there is hardly any upslope contributing area. Thus, $R$ in the water balance equation can be assumed to be dominated by landscape-scale subsurface runoff leaving the headwater area. This runoff

can finally be expected to enter nearby rivers that drain the mountain range. Streamflow time series measured at the gauge Chamerau (Sect. 2), converted to specific runoff in mm water equivalent, are thus used as the basis for quantifying the runoff component in the water balance of Eq. (3).

$$\mathrm{d}S/\mathrm{d}t = s \cdot \left(\mathrm{d}g/\mathrm{d}t - v \cdot \mathrm{d}g_{glob}/\mathrm{d}t\right) = u \cdot P - a \cdot E_{ref} - c \cdot R \qquad (3)$$

Although the problem in Eq. (3) is linear, the parameters are not linearly independent. The optimization problem was therefore solved by introducing additional parameter constraints and applying a non-linear optimization approach using the interior-point algorithm (Matlab R2015b). To evaluate the statistical match of the daily water storage time series of the left and the right side of Eq. (3) we apply the Kling-Gupta efficiency (KGE) (Gupta et al., 2009) and use KGE as the performance criterion

to be maximized during optimization. The optimization was performed by adjusting the parameters $a$, $c$, $u$ and $v$ in Eq. (3) without explicitly enforcing the closure of the water budget over the analysis period. Setting the closure of the water budget as a constraint would imply that the gravity residuals were not affected by errors, which apparently cannot be assumed so that imperfect instrumental (drift and steps) and gravity (e.g. atmosphere) corrections would directly propagate into the estimated parameter. Table 2 shows the a-priori defined parameter ranges. In case of evapotranspiration, the factor $a$ converts $E_{ref}$ into

actual evapotranspiration E and, hence, $a$ can vary between 0 and 1. The precipitation factor $u$ accounts for inaccuracies in the under-catch correction. The lower bound is set to 0.9 which approximates precipitation without under-catch correction, the upper bound is set to 1.1 to account for a possible underestimation of the correction. The runoff factor $c$ can be interpreted as a correction for conceptual mismatches (e.g., river runoff at the large catchment scale may not be fully equivalent to the subsurface runoff component considered at the gravimeter scale) and for conversion errors (e.g., inaccurate catchment area).

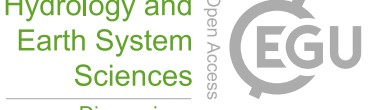



The lower and upper bounds of $c$ were initially set to allow for a maximum change of specific runoff by 10% in both directions. These bounds were further kept for the analysis as they were never reached during the optimization process. In addition, an uncertainty factor $v$ for the contribution of the large-scale hydrological gravity effect on the left-hand side of the equation is included. Given large difference in estimates of this gravity effect when different global hydrological models are used (e.g.,

Mikolaj et al., 2016), varying $v$ during optimization allows for accounting for this uncertainty. The parameter bounds of $v$ were derived from the minimum and maximum multiplicative factors that were needed to convert the large-scale hydrological gravity effect of each of the four different hydrological models used here to the mean effect of the four models ($\mathrm{d}g_{glob}/\mathrm{d}t$).

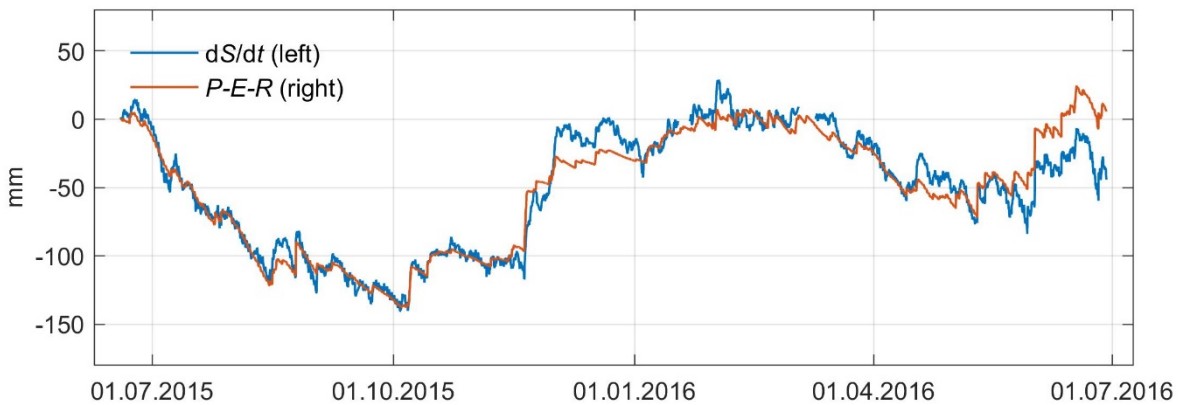

**Figure 10: Time series of water storage at Wettzell (as a deviation from the initial storage value at the beginning of the study period, arbitrarily set to 0) after optimization with iGrav006 gravity-based storage on the left (blue line) and the optimized right side (red line) of the water balance equation (Eq. (3)).**

The optimization (period 19 June 2015 – 29 June 2016) resulted in a very high Kling-Gupta efficiency of 0.98, mainly due to

the good fit of the dominant seasonal storage variations and no considerable bias (Figure 10). An exception with a larger bias is towards the end of the study period where major steps in the gravity time series had to be removed manually prior to optimization (see Chapter 3.2). The correlation coefficient between the optimized storage time series on the left and right side of Eq. (3) is 0.95, the mean difference is 0.02 mm, and the standard deviation is 13 mm. The optimized parameter values are listed in Table 2. The runoff factor $c = 1.08$ means that runoff from the gravimeter footprint is slightly increased relative to

measured streamflow within the optimization procedure. Reasons may include unaccounted groundwater discharge in the valley bottom of the river gauging station or higher than average gradients for subsurface flow processes in the iGrav headwater region. The precipitation factor $u = 1.00$ indicates that the under-catch correction by Richter, 1995 is reasonable for this site so that no further adjustment of precipitation volumes by the optimization approach is required. The evapotranspiration factor $a = 0.69$ means that the actual evapotranspiration of the landscape around the Wettzell observatory is about 69% of the

(potential) grass reference evapotranspiration. This in turn shows that the hydrological system was at least partly water limited for evapotranspiration during the study period, in spite of its location in a mountainous humid temperate climate regime. One





contributing factor is that the period included an exceptional drought in summer 2015 that hit in particular Southern Germany and the area of the Czech Republic (Laaha et al., 2016) where Wettzell is part of. To assess the validity of the latter two factors, we compared them to values derived in a completely independent way from the lysimeter time series at the Wettzell observatory. The corresponding lysimeter-based factor $u*$ was computed as the ratio between lysimeter precipitation

(determined from its mass increase during rainfall events) and gauge precipitation corrected for under-catch. The factor $a*$ is the ratio between the lysimeter-based actual evapotranspiration and the grass reference evapotranspiration. The lysimeter-based factors $u*$ and $a*$ for the study period are 1.02 and 0.68, respectively, and thus very close to the gravity-based optimization results (Table 2). These results show that the superconducting gravimeter in the field deployment is very well suited for quantifying the annual constituents of the water balance equation. For the 1-year period (mid June 2015 – mid June

2016) in Wettzell, the annual values derived from the gravity-constraint approach for $P, E, R$ and for gravity-based $dS/dt$ were 829, 412, 394 and -8 mm, respectively. Thus, in total, the mismatch of $P - E - R$ versus $dS/dt$ amounted to 31 mm. This value, corresponding to about 4% of annual precipitation, can be considered as the error in closing the water balance at the annual scale by the gravity approach.

## 4.3 Resolving water balance components – daily scale

Quantifying actual evapotranspiration is of particular interest due to the lack of other direct observation techniques at the stand or landscape scales, with the exception of the eddy covariance method (Baldocchi et al., 1988). The questions arises whether the gravity-based water balance approach presented above can be used to quantify $E$ over shorter periods of time, ideally on a daily basis to assess the landscape $E$ response to changing conditions in terms of meteorological drivers, water availability and the physiological status of plants. The main obstacles are instrumental issues such as the spurious temperature effects on the

gravity time series mentioned above, and a deficient correction of non-hydrological effects, e.g. tides or ocean loading. Van Camp et al. (2016a) recently needed to stack the gravity time series over several periods without rainfall to isolate a mean daily value for $E$. For the optimization presented here, we thus solve the water balance within moving windows of several days in length instead of using day to day differences. Different from the optimization described above, only the evapotranspiration factor $a$ is adjusted (here in a time-variable way, i.e., for each moving window) while the other three factors are taken as

constant values as derived in the optimization before (Table 2). Prior to the final optimization we test the following question: Does this method of scaling $E_{ref}$ within in a moving window approach result in actual evapotranspiration at the daily scale? To answer this question, we run a moving window optimization for the time-varying factor $a$ using the $E_{ref}$ time series as input and the observed time series of lysimeter $E$ as the target. Then, daily $E$ is estimated by multiplying $a$ with $E_{ref}$ for the central day of the considered window. The difference between estimated and actual lysimeter evapotranspiration is the error of the

method. The root mean square error (RMSE) for windows of 9, 11 and 13 days in length is 0.16, 0.17 and 0.18 mm d$^{-1}$, respectively. A further increase of the window length gradually degrades the accuracy. This shows that the method itself is indeed capable of resolving daily evapotranspiration rates with sub-mm accuracy when setting the window to a reasonable length.





The results of the final time-varying optimization of *a* in Eq. (3) with fixed factors for precipitation, streamflow and gravity-based storage change are shown in Figure 11 for a moving window length of 11 days. The estimated actual evapotranspiration fits the lysimeter observations well, both with regard to the magnitude of daily $E$ rates as well as their temporal variations. In particular during the summer season (July-August), estimated $E$ usually is considerably smaller than $E_{ref}$, as also indicated by comparatively small values of *a*. This demonstrates the water-limited state of the hydrological system during this period. $E$ (both from gravity and from lysimeter) tends to be close to $E_{ref}$ during the autumn season with overall much smaller daily $E$ rates.

An exception of the overall good performance is the period between August 22 and August 30 where estimated daily E rates are always equal to $E_{ref}$ and systematically overestimate observed lysimeter $E$ (Figure 11). Within this time period, the RMSE of estimated versus lysimeter evapotranspiration is 0.92 mm d⁻¹, while it is 0.42 mm d⁻¹ outside of this interval. This discrepancy is related to a strong decrease of storage as given by the gravimeter in this period, with an average of 4 mm d⁻¹. Unrealistically high runoff rates of more than 1 mm d⁻¹ would be required to match these storage change rates with the observed $E$ of the lysimeter which is in the order of 2.5 mm d⁻¹. Hence, the apparent storage decrease is probably related to insufficient correction of gravity effects in the iGrav time series that are not related to local water storage. However, the strong storage decrease is neither reduced by the use of different global model for the large-scale hydrological effects, nor by the inclusion of non-tidal ocean loading effects or by a change of gravimeter drift rate. We furthermore checked possible atmospheric effects, motivated by the fact that the atmosphere contains the evaporated mass which affects the gravity with opposite sing due to the change of its position relative to gravimeter sensor (from the soil below to the atmosphere above the instrument). The applied 3D Atmacs correction was replaced by (i) a 3D correction using the ERA Interim model (Mikolaj et al., 2016) and (ii) a simple approach using in-situ observed air pressure solely instead of a full 3D field. None of these changes, however, affected the $E$ estimations in this dry period in a considerable way so that the reason for the discrepancies remains unresolved. Large-scale atmospheric mass increase by evaporated water during the strong evapotranspiration period that could not be included in atmospheric models nor in the local air pressure observations remain an unproven hypothesis. It should, however, also be noted that the observation data taken here from the lysimeter are very different in spatial scale than the $E$ effect seen by the gravimeter. Differences in $E$ between the lysimeter and the landscape scale may thus also contribute to the differences in the time series. This is not necessarily a limited performance of the gravimetry-based approach, but an expression of its better suitability for quantifying landscape-scale hydrological dynamics.





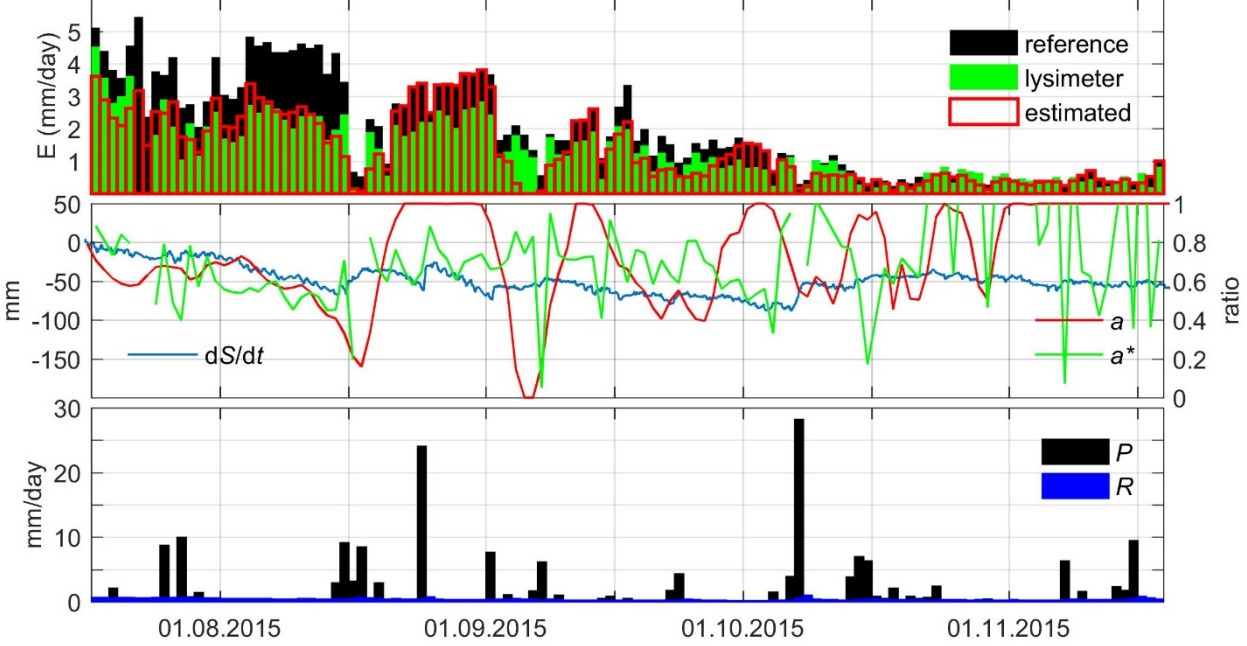

**Figure 11: Upper panel: Comparison of (Penman-Montheith grass) reference evapotranspiration, observed lysimeter-based actual evapotranspiration (zero values of lysimeter $E$ indicate missing data), and estimated actual evapotranspiration (from gravity-constraint optimization). Middle panel: Gravity-based storage anomaly relative to 10.07.2016, the optimized factor $a$ of the water balance equation (ratio of gravity-based $E$ to $E_{ref}$) and the factor $a^*$ based on the lysimeter time series (ratio of lysimeter-based $E$ to $E_{ref}$). Lower panel: Input time series of precipitation and runoff.**

## 5 Conclusions

Observing mass budgets in environmental systems is a fundamental challenge and rarely possible in a comprehensive way due to the multitude and complexity of pools and fluxes involved. This also applies to the hydrological cycle when it comes to monitoring the water balance at spatial scales in between the point and the river basin scale. With the field observation technique presented here, based on continuous gravimetry with a superconducting gravimeter (SG), an unprecedented means of directly monitoring the water balance at the 100 – 1000 meter scale becomes available.

By deploying a SG in a small field enclosure, we demonstrate for the first time that a continuous and stable outdoor operation of a SG is feasible for a long time (here more than one year) under humid environmental conditions with marked daily and seasonal temperature variations. At the same time, the quality of the gravity time series does not degrade in comparison to the standard SG deployment under controlled conditions in observatory buildings. The field enclosure design proves to shield the instrument sufficiently well from temperature variations, wind pressure or other environmental effects that may cause vibrations, instrument tilt or other spurious effects. Thus, the tiny hydrology-induced gravity signal of interest is only marginally obscured by instrument noise. We show that the deployment of the SG in a field enclosure conveys other advantages



relative to the existing SGs in observatory buildings. First, being spatially closer to the signal of interest, the field SG is more sensitive to local water storage variations and it is not affected by unnatural and usually unknown storage variations below a building. Secondly, we demonstrate that the gravity residual time series of the field SG are a direct expression of the total water mass change in the surroundings of the instrument, almost independent of the depth below the terrain surface where the

storage changes occur. Thus, with the field SG, we present the first continuous and integrative monitoring technique of the landscape water budget. It should be noted that this conclusion applies if the storage changes occur within the full integration radius of the SG (few hundreds of meters) with low horizontal heterogeneity. In contrast, spatially constrained water storage changes, such as in the case of an artificial sprinkling experiment in the vicinity of the gravimeter, result in a dedicated sensitivity of the SG to the depth of the storage change, and can be used for identifying the infiltration process (Reich et al., in

preparation: "Field-scale infiltration dynamics inferred from continuous gravity monitoring during sprinkling experiments"). With the gravity monitoring system presented here, we show that the annual water balance can be closed within 4% of annual precipitation. This error results from imperfect reduction of mass signals other than the local hydrological ones in the gravity time series, and of instrumental effects such as drift and steps. We provide a framework to quantify the individual components of the water balance from the gravity observations at annual to daily time scales. Notably, expanding the potential as pointed

out by Van Camp et al. (2016a), we demonstrate the value of the field SG as a technique for assessing actual evpotranspiration. The accuracy of the approach when evaluated against daily ET rates from lysimeter time series was in the order of 0.5 to 1 mm d$^{-1}$, but the different spatial footprints of both methods limit their direct comparability. In turn, if reasonable data of actual ET were available from other observation techniques such as the eddy covariance method, a collocated field SG will offer a unique means of estimating subsurface runoff via the gravity-based water balance approach presented here.

From a practical perspective, and compared to other hydrological field monitoring techniques, the widespread and flexible deployment of the field SG system proposed here may currently be hampered by the need of a solid gravimeter fundament, the weight and complexity of the monitoring system, and by its costs. Nevertheless, this study lays out the potential of high-precision gravimetry in the field as a non-invasive observation method that fills gaps in the spectrum of existing hydrological and hydro-geophysical methods with respect to the target observables and the spatial scale to be captured. Ongoing

technological development towards smaller gravimeters including alternative techniques for high-precision gravity measurements such as quantum gravimeters give prospect for a much broader future application of the hydro-gravimetric principles developed here.




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



**Table 1: Parameters used at the observatory Wettzell to correct precipitation ($P$) for under-catch by applying the equation $P_{cor} = P + bP^\varepsilon$. Liquid precipitation between March and November is treated as summer rain, liquid precipitation during the other months as winter rain.**

| Precipitation type | $\varepsilon$ | $b$ |
| --- | --- | --- |
| Liquid/summer | 0.380 | 0.280 |
| Liquid/winter | 0.460 | 0.240 |
| Mixed | 0.550 | 0.305 |
| Snow | 0.820 | 0.330 |





**Table 2: Parameters of the water balance equation adjusted during optimization (see text for detailed explanations)**

| Parameter scope | Name | Parameter range | Optimized value | Independent value from lysimeter |
|---|---|---|---|---|
| Evapotranspiration | $a$ | 0.00 – 1.00 | 0.69 | 0.68 |
| Precipitation | $u$ | 0.90 – 1.10 | 1.00 | 1.02 |
| Runoff | $c$ | 0.90 – 1.10 | 1.08 | - |
| Global hydrological gravity effect | $v$ | 0.49 – 1.28 | 1.28 | - |