# Peer review of "Landscape-scale water balance monitoring with an iGrav superconducting gravimeter in a field enclosure"

_Hydrology and Earth System Sciences, 2017_

## Referee Comment (RC1) · Dr. Jahr (Referee) · 28 Mar 2017

Reviewer comments on:

"Landscape-scale water balance monitoring with an iGrav superconducting gravimeter in a field enclosure"

by Güntner et al.

This is a really excellent paper. We need exactly such investigations for a better understanding of hydrological induced gravity signals. The quality of content and form is high and I have only some small suggestions. (= minor revision)

1) page 1, line 16

I understand this sentence because I've read the whole paper. But may be it would be good to add also a sentence about the installation - then it's more clear.

2) p2, l11 - l13

different scales are introduces like "...regional to continental scale" or "landscape scale". But how large is a landscape scale? A quantitative hint would be good!

3) p3, l6

sometimes the citation is "..., Van Camp et al., 2016a, presented..."

but more usual to cite: "..., Van Camp et al. (2016a) presented....

4) figure 1

In the DEM region seems to be some blue text? ...but it's to small and also the scale in the left corner is hard to read.

Not everybody knows where Wettzell is located in Germany. It would be good to add a small map og Germany (in the right corner) with the marked location

5) p5, l11

Richter (1995)

6) p6, l23

how constant is the temperature inside the box?

7) p6, l24+25

the 2 m²: is it experience or modelled or....?

8) p7, l16

here the unit is separated from one line to the next. It should be used as one word!

9) p7, l24+25

do you have any ideas for the reasons of the steps? ...and how big are these steps?

10) p8, l10

is the resulted drift also caused by insufficient step correction? ..or not?

11) p9, l4

stabilized temperature: ...in which range? how stable? (it would be better to argue quantitaively)

12) p11, figure caption

we need a citation after (NLNM) ...Petersen 1993?

13) p11, l11

Bonatz (1967)

14) p12, l5-10

can you say anything about tilt effects of the pillar caused by temperature changes (due to water close to the pillar?)

15) p17, formular

all variables should be explained. This is done six lines later. It would be good if this would be close to the formular!

16) p18, l22

Richter (1995)

17) p22, l16-17

no separation of number and the unit

Reference list:

p25, l17

it should be: "New Mexico, 1993.

---

## Referee Comment (RC2) · Anonymous Referee #2 · 30 Mar 2017

I have reviewed the paper "Landscape-scale water balance monitoring with an iGrav superconducting gravimeter in a field enclosure," by A. Güntner et al., which presents a new capability for deploying gravimeters in a field setting. The paper nicely shows how the gravity response to infiltration becomes more linear as a direct result of the sensitivity allowed by the pillar/enclosure. By demonstrating a novel application that could have wide applicability, I find the paper deserving of publication in HESS-D.

I found no major errors or concerns with the paper. A number of minor revisions are included below.

I suggest the authors cite the Wilson, et al. paper that describes an earlier attempt at using the superconducting gravimeter as a field instrument. Although the present effort using an iGrav is an advance over the Wilson field enclosure, it is worthwhile to show that others have also considered ways to improve gravity data collection for field studies. [Wilson, C.R., B.R. Scanlon, J. Sharp, L. Longuevergne and H. Wu. 2012. Field test of the superconducting gravimeter as a hydrologic sensor. *Ground water* 50, no. 3: 442–9.]

For studies of aquifer recharge (i.e., infiltration that has entered the aquifer), near-surface soil moisture is "noise" that must be estimated and removed in order to identify recharge. In that case, the decreased near-surface sensitivity afforded by the umbrella effect beneath a large building/enclosure is an advantage, and a small pillar/field enclosure would decrease the ability to identify recharge. It may be worth noting the iGrav/tall pillar combination isn't useful for every hydrologic investigation.

Lines 16-18: I find the description of the instruments confusing. It might be clearer just to state relative gravimeters measure relative gravity differences (between stations or over time). It could be just as accurately said that absolute meters also measure the impact of the resulting force changes on a test mass. Also consider explaining what you mean by "continuous" (including the ~45 second filter on relative meters).

Line 17: the magnitude of the vertical gravity vector (as written, the implication is the meter measures the 3-d vector).

Line 22: It would be helpful to state explicitly that spring and superconducting meters are relative meters.

Line 26: Usage of "time-lapse" is vague; in the previous paragraph you use it to mean "periodically repeated", but some of the references cited here use SG (continuous) data.

Line 33: Consideration of these effects (tides, loading) are important for all gravity studies, not just those using SGs. Lines 33-36 could be deleted to help the paragraph flow better.

P.3, Line 23: It might be useful to indicate this study is concerned with near-surface soil-moisture changes, which weren't relevant or observed in the Kennedy (2014) paper, which also demonstrated the iGrav in a field enclosure.

Line 28: gneiss should be lower case.

Figure 1: What is the coordinate system? Suggest changing "gauging station" to "streamgauging station".

Page 6, line 3: Usually "on the one hand" and "the other hand" are used to contrast two items. In this case the two enclosures are just two parts of the system, not contrasting parts.

Page 6: It would be helpful to indicate the power requirement of the iGrav (and that it requires AC (line) power, if that is the case).

Page 7, line 20: Suggest including the URL for the Atmacs service (I wasn't previously familiar with it).

Page 8, line 5: meaning of "could not be carried precisely enough" is not clear. What are the implications for determining drift for other field studies, where a lysimeter is not present? If drift cannot be determined using the abundance of gravity meters available at Weitzell, is there any hope for other field sites? Can you make any inferences regarding the linearity of drift (perhaps there are other periods with identical soil moisture/groundwater levels)?

Would it be possible to show a cutaway schematic of the field enclosure? From the description the annular air space and configuration of cooling grills isn't entirely clear.

Page 9, line 10: Can you indicate the temperature at which PCB effects began to be seen? I don't see anywhere the maximum temperatures observed at Wetzell, but I'm guessing it's relatively mild. Would these temperature effects be expected at pretty much every installation? Can you comment as to whether these temperature effects have been seen in other (non-iGFE) iGrav installations? Perhaps it would be useful to include a table showing the range of environmental conditions (temperature, wind speed, humidity, etc.) under which the iGrav operated with satisfactory results.

Page 10, line 16: Cold head = cryocooler, correct? Consistent terminology will help tie in with the discussion on page 6.

P.11, line 10: suggest replacing perpendicular with horizontal.

Figure 6: A useful figure, as it's not intuitive the departure from an infinite slab would be so great for a 1m high pillar. I believe the purple dashed line (infinity symbol in the legend) is for the infinite radius case, not infinite height, as shown? The gravitational attraction of a finite-radius cylinder would approach zero if the gravimeter were at infinite height.

p. 13, line 11: I think the 2.17 and 0.15 numbers are reversed, the iGrav should have a larger gravity effect?

p.13: Can you talk about the difference in sensitivity to storage change at the water table for the 2 sites? Is it equal?

Figure 8: Does the "double-peak" behavior warrant explanation? I assume the increase at distances between 500 and 3000 radius comes from valleys surrounding the site?

Its obvious, but it may be helpful to say explicitly that the gravity signal never decreases as the wetting front moves deeper – it can only decrease if soil water is evaporated, or somehow moves laterally (usually as groundwater, after recharge). That feature makes the method equally useful for thick unsaturated zones.

p 15, line 17: what is the soil porosity? The precip flux and Ks are both very high, although the response in figure 9 would be similar for a broad range of parameters.

p.18, line 5: there are a few references where e.g. is used but not needed

Figure 10: It is worth noting in the caption or on the figure that only the first 10 days are used for optimization. I would move "after optimization" to later in the sentence, so it is obvious the blue line is data. Maybe delete "(as a deviation from…)"

It seems that P-E-R misses some of the high-frequency behavior in the gravity residuals. Can you comment? Is it because s is slightly too high for near-surface soil moisture change? I.e., when converting

from dg to ds, there is a slightly non-linear effect that would make s, as shown in figure 10, slightly less dynamic.

p.19, line 19: Worth noting there are many factors (including all those mentioned), with the same 24-hour period as ET.

line 23: State explicitly why day-to-day ET can't be determined directly from the data (i.e., rearrange eq. 3 to solve for (Eref – c) without stacking or a moving window. You mention the main obstacles earlier, but it's not obvious these are the exact (and only) reasons it can't be done. line 30 would indicate the estimate improves as the window gets even shorter than 9 days?

p. 20, line 17: sign not sing

I find the discussion about the E discrepancy very interesting, as it's not obvious to me how a gravimeter at the land surface responds to ET – to decrease gravity seems to require advecting the evaporated water away from the region of sensitivity.

p.22, line 9: I'm unclear what a "dedicated sensitivity" is. Is it appropriate to cite a paper in preparation?

line 21: I'm not sure "fundament" is widely known (it's unfamiliar to me). Also, the significant limitation of requiring AC power.

---

## Author Comment (AC1) · 19 May 2017

We are grateful to both reviewers for their positive and constructive reviews of our manuscript. We will incorporate their helpful suggestions in a revised version of the manuscript. In the following, we give a point by point response (denoted by R1, R2, ...) to all reviewer comments.

Reviewer 1:

This is a really excellent paper. We need exactly such investigations for a better understanding of hydrological induced gravity signals. The quality of content and form is high and I have only some small suggestions. (= minor revision)

[Figure]

1) page 1, line 16: I understand this sentence because I've read the whole paper. But may be it would be good to add also a sentence about the installation - then it's more clear.

R1) It is not clear to which sentence the reviewer refers to. Adding a sentence on the installation at this point of the abstract does not seem helpful to us.

2) p2, l11 - l13: different scales are introduces like "...regional to continental scale" or "landscape scale". But how large is a landscape scale? A quantitative hint would be good!

R2) We specify the landscape scale following as the scale in the range of several hundreds to thousands of meters and add this to the revised manuscript.

3) p3, l6: sometimes the citation is "..., Van Camp et al., 2016a, presented..." but more usual to cite: "..., Van Camp et al. (2016a) presented....

R3) Corrected according to the HESS standards in the revised version of the manuscript.

4) figure 1: In the DEM region seems to be some blue text? ...but it's to small and also the scale in the left corner is hard to read. Not everybody knows where Wettzell is located in Germany. It would be good to add a small map og Germany (in the right corner) with the marked location.

R4) The blue text in DEM denotes the names of the major rivers in the region. We will provide better readable text in the revised version of the figure, together with an overview map Germany to illustrate the location of Wettzell, following the reviewer's suggestion.

5) p5, l11: Richter (1995)

R5) Corrected.

6) p6, l23: how constant is the temperature inside the box?
R6) The performance of the air conditioning system with regard to temperature stability is discussed in section 3.3. Figure 4 provides data of the temperature inside the box based on the electronics board temperature.

7) p6, l24+25: the 2 m$^2$: is it experience or modelled or....?

R7: The footprint of the iBox is 1 m$^2$. We specified the revised manuscript with this value. The statement that it has a negligible umbrella gravity effect at a distance of 12 meters from the iGrav is based on sensitivity calculations: a hypothetical water storage change of 100 mm on the footprint of the iBox causes a gravity change smaller than 0.01 nm/s2.

8) p7, l16: here the unit is separated from one line to the next. It should be used as one word!

R8: Corrected.

9) p7, l24+25: do you have any ideas for the reasons of the steps? ...and how big are these steps?

R9: The total absolute value of all removed steps is 1230.4 nm/s2, while the amplitude of the biggest step reached 654.7 nm/s2. Five steps are related to maintenance (e.g., cold-head exchange and cleaning), two have unknown source, and the remaining two steps are caused by power surge and iGrav software upgrade. This information is added to the revised manuscript.

10) p8, l10: is the resulted drift also caused by insufficient step correction? ..or not?

R10: The drift can be indeed partially affected by insufficient step correction. This comment has been added to the revised manuscript. Nevertheless, the major part of the drift has still to be attributed to mostly unexplained instrumental effects, depending among others on the initialization procedure, the temperature control of the sensor, gas adsorption and desorption from the sphere (Goodkind, 1999).

11) p9, l4: stabilized temperature: ...in which range? how stable? (it would be better to argue quantitatively)

R11: Section 3.3 and Figure 4 give an extensive and quantitative explanation on the temperature variations before and after adjusting the ventilation system in the enclosure. In the revised manuscript, we point directly to this on page 9, line 4, as cited by the reviewer.

12) p11, figure caption: we need a citation after (NLNM) ...Petersen 1993?

R12: Corrected.

13) p11, l11: Bonatz (1967)

R13: Corrected.

14) p12, l5-10: can you say anything about tilt effects of the pillar caused by temperature changes (due to water close to the pillar?)

R14: Tilting of the pillar due to environmental temperature effects and insolation cannot be excluded. In the time series of the tilt compensation system, we indeed do see diurnal signals during sunny summer days. Nevertheless, all tilt effects could be efficiently compensated for by the tilt compensation system of the gravimeter, as could be proved by an equilibrated tilt balance time series. This is discussed at the end of section 3.3 of the manuscript.

15) p17, formular: all variables should be explained. This is done six lines later. It would be good if this would be close to the formular!

R15: The explanation of the variables used in Eq. 3 covers four paragraphs before and after the equation, as for clarity of the approach we considered a somewhat longer explanation in plain text to be appropriate. Following the reviewer's recommendation, we add lines with short descriptions of all variables directly after the equation in the revised version of the manuscript.

16) p18, l22: Richter (1995)

R16: corrected.

17) p22, l16-17: no separation of number and the unit

R17: corrected.

18) Reference list: p25, l17: it should be: "New Mexico, 1993

R18: corrected

Reviewer 2:

I have reviewed the paper "Landscape-scale water balance monitoring with an iGrav superconducting gravimeter in a field enclosure," by A. Güntner et al., which presents a new capability for deploying gravimeters in a field setting. The paper nicely shows how the gravity response to infiltration becomes more linear as a direct result of the sensitivity allowed by the pillar/enclosure. By demonstrating a novel application that could have wide applicability, I find the paper deserving of publication in HESS-D.

I found no major errors or concerns with the paper. A number of minor revisions are included below.

I suggest the authors cite the Wilson, et al. paper that describes an earlier attempt at using the superconducting gravimeter as a field instrument. Although the present effort using an iGrav is an advance over the Wilson field enclosure, it is worthwhile to show that others have also considered ways to improve gravity data collection for field studies. [Wilson, C.R., B.R. Scanlon, J. Sharp, L. Longuevergne and H. Wu. 2012. Field test of the superconducting gravimeter as a hydrologic sensor. Ground water 50, no. 3: 442–9.]

R1: We thank the reviewer for reminding us of this earlier study and fully agree that it should be cited in our manuscript. We refer to Wilson et al. 2012 and Hector et al. 2014 as previous attempts to use superconducting gravimeters as hydrological sensors

in the field in the revised version of the manuscript.

For studies of aquifer recharge (i.e., infiltration that has entered the aquifer), near-surface soil moisture is "noise" that must be estimated and removed in order to identify recharge. In that case, the decreased near-surface sensitivity afforded by the umbrella effect beneath a large building/enclosure is an advantage, and a small pillar/field enclosure would decrease the ability to identify recharge. It may be worth noting the iGrav/tall pillar combination isn't useful for every hydrologic investigation.

R2: We completely agree and add a short note on the (limited) value of the iGrav/tall pillar setup following the reviewer's reasoning in the conclusions part of the revised manuscript.

Lines 16-18 (page 2): I find the description of the instruments confusing. It might be clearer just to state relative gravimeters measure relative gravity differences (between stations or over time). It could be just as accurately said that absolute meters also measure the impact of the resulting force changes on a test mass. Also consider explaining what you mean by "continuous" (including the ∼45 second filter on relative meters).

R3: Corrected and specified according to the reviewer's suggestion. The time delay caused by the filter and the feedback loop, ranging from 5 to 45 seconds depending on the used electronics, is corrected during data pre-processing. This aspect is not explicitly explained in the present paragraph because it does not appear to be essential for the general overview at this point.

Line 17: the magnitude of the vertical gravity vector (as written, the implication is the meter measures the 3-d vector).

R4: The magnitude of the gravity vector defines the direction of the vertical, independently of the choice of the coordinate system. A gravimeter is precisely orientated in this direction. The formulation 'vertical' gravity vector would hence be a duplication or

would describe the special case where the coordinate system is orientated in such a way that magnitude of the vector is identical with the vertical (third) component of the vector.

Line 22: It would be helpful to state explicitly that spring and superconducting meters are relative meters.

R5: Added to the revised manuscript.

Line 26: Usage of "time-lapse" is vague; in the previous paragraph you use it to mean "periodically repeated", but some of the references cited here use SG (continuous) data.

R6: We agree, this is confusing. We refer to time-lapse as periodically repeated measurements only. Thus, in this part, we removed the reference by Creutzfeldt et al. (2010) that used continuous SGs observations.

Line 33: Consideration of these effects (tides, loading) are important for all gravity studies, not just those using SGs. Lines 33-36 could be deleted to help the paragraph flow better.

R7: As the audience of the paper, among others, are hydrologists that may not be familiar with gravimetry, it should be important to note that these non-hydrological signal components of gravity time series have to be removed prior to hydrological interpretation. Thus, we suggest to keep these lines in the manuscript.

P.3, Line 23: It might be useful to indicate this study is concerned with near-surface soil-moisture changes, which weren't relevant or observed in the Kennedy (2014) paper, which also demonstrated the iGrav in a field enclosure.

R8: In the present study, we are interested in total water storage variations, including both near-surface and deeper vadose and saturated zone storage changes. It is true, though, that (natural rain and ET-caused) near-surface soil moisture changes were non-existent for the Kennedy et al. (2014) study. We extended the sentence accordingly.

Line 28: gneiss should be lower case.

R9: Corrected.

Figure 1: What is the coordinate system? Suggest changing "gauging station" to "streamgauging station".

R10: Figure 1 will be modified accordingly. The coordinate system is UTM.

Page 6, line 3: Usually "on the one hand" and "the other hand" are used to contrast two items. In this case the two enclosures are just two parts of the system, not contrasting parts.

R11: Corrected.

Page 6: It would be helpful to indicate the power requirement of the iGrav (and that it requires AC (line) power, if that is the case).

R12: Sentence on power requirements added to the revised manuscript.

Page 7, line 20: Suggest including the URL for the Atmacs service (I wasn't previously familiar with it).

R13: URL added.

Page 8, line 5: meaning of "could not be carried precisely enough" is not clear. What are the implications for determining drift for other field studies, where a lysimeter is not present? If drift cannot be determined using the abundance of gravity meters available at Weitzell, is there any hope for other field sites? Can you make any inferences regarding the linearity of drift (perhaps there are other periods with identical soil moisture/groundwater levels)?

R14: With the statement that measurements with an absolute gravimeter could not be carried precisely enough for drift determination, we refer to the fact that operating a FG5 directly adjacent (within the next meters) to the iGrav in its field enclosure was not

reasonably possible because there was no housing for the FG5. Thus, collocated FG5 observations were not possible at all, and even A10 measurements were not feasible due to the absence of an additional monument, despite the limited accuracy. At other field sites, drift estimation will be a challenge, too. One possible approach is to assume the drift rate of a particular iGrav to be constant when transported to the new site in cold conditions, so that the previously determined drift at an observatory site based on FG5 measurements can be taken. Alternatively, in future, the deployment of absolute quantum gravimeters based on atom interferometry might be an option with sufficient accuracy for SG drift estimation for field applications. The linearity of the iGrav006 drift could not be further assessed as the length of the time series (approx. one year) did not allow for identification of other periods with identical water storage conditions.

Would it be possible to show a cutaway schematic of the field enclosure? From the description the annular air space and configuration of cooling grills isn't entirely clear.

R15: We decided not to include a scheme with the details of the field enclosure design in the manuscript as the enclosure used in this study was still a prototype version that underwent further modifications by the manufacturer GWR during the last years. Thus, once commercially available in its final design, GWR is expected to provide a schematic of its configuration.

Page 9, line 10: Can you indicate the temperature at which PCB effects began to be seen? I don't see anywhere the maximum temperatures observed at Wetzell, but I'm guessing it's relatively mild. Would these temperature effects be expected at pretty much every installation? Can you comment as to whether these temperature effects have been seen in other (non-iGFE) iGrav installations? Perhaps it would be useful to include a table showing the range of environmental conditions (temperature, wind speed, humidity, etc.) under which the iGrav operated with satisfactory results.

R16: The diurnal temperature effects were a problem of the initial enclosure design. As mentioned in section 3.3, with the installation of fans to circulate the air in the enclosure

we could solve this issue. It can be expected to be solved also for future installations under similar environmental conditions. To this end, as suggested by the reviewer, we add the range of meteorological conditions under which the iGrav could be successfully operated at Wettzell in the revised manuscript in section 3.3.

Page 10, line 16: Cold head = cryocooler, correct? Consistent terminology will help tie in with the discussion on page 6.

R17: yes, terminology unified throughout the manuscript (cryocooler).

P.11, line 10: suggest replacing perpendicular with horizontal.

R18: Modified.

Figure 6: A useful figure, as it's not intuitive the departure from an infinite slab would be so great for a 1m high pillar. I believe the purple dashed line (infinity symbol in the legend) is for the infinite radius case, not infinite height, as shown? The gravitational attraction of a finite-radius cylinder would approach zero if the gravimeter were at infinite height.

R19: Correct. Figure and figure caption adjusted in the revised version of the manuscript.

p. 13, line 11: I think the 2.17 and 0.15 numbers are reversed, the iGrav should have a larger gravity effect?

R20: No, the numbers are correct in the way they are given. The numbers are for a hypothetical gravity effect, i.e., what was the gravity effect if a certain water storage change occurred below the gravimeter building and below the iGrav pillar, respectively. Because of the larger size of the building, this gravity effect would be much larger for the SG than for the iGrav. For natural conditions, in turn, the gravity effect is indeed larger for the iGrav, as explained in the next sentence.

p.13: Can you talk about the difference in sensitivity to storage change at the water

table for the 2 sites? Is it equal?

R21: As mentioned towards the end of this paragraph, if water storage variations occur in larger soil depths (such as water table variations), i.e., at a larger distance for the gravimeter, effects of height differences of the sensor relative to the zone where storage variations occur becomes smaller. In addition, as the water table at the site does not follow the terrain surface in a parallel way, but it is deeper below the surface at higher topographic positions, we can expect less compensation effect by groundwater at the SG30 site than by soil moisture variations. Thus, overall, the signal by groundwater variations at the two sites can be expected to be more similar than by unsaturated zone moisture variations.

Figure 8: Does the "double-peak" behavior warrant explanation? I assume the increase at distances between 500 and 3000 radius comes from valleys surrounding the site?

R22: Yes, exactly, the second step/increase in the sensitivity curves in Figure 8 is due to topographic effects, i.e., valleys in the vicinity of the observatory which itself is located on a ridge. We added two sentences for explanation of this behaviour to the manuscript.

Its obvious, but it may be helpful to say explicitly that the gravity signal never decreases as the wetting front moves deeper – it can only decrease if soil water is evaporated, or somehow moves laterally (usually as groundwater, after recharge). That feature makes the method equally useful for thick unsaturated zones.

R23: We fully agree. To highlight this feature, we added half a sentence to the conclusions, stressing the value of the approach for deep groundwater and/or a thick unsaturated zone.

p 15, line 17: what is the soil porosity? The precip flux and Ks are both very high, although the response in figure 9 would be similar for a broad range of parameters.

R24: The porosity assumed in the modelling experiment was 37 Vol% (added to the

revised manuscript). Indeed, the experiment is a rather arbitrary illustration of an effect that can similarly by expected for a broad range of conditions.

p.18, line 5: there are a few references where e.g. is used but not needed

R25: Corrected.

Figure 10: It is worth noting in the caption or on the figure that only the first 10 days are used for optimization. I would move "after optimization" to later in the sentence, so it is obvious the blue line is data. Maybe delete "(as a deviation from...)"

R26: As mentioned in the text, optimization was done for the entire period of about one year. Nevertheless, according to the reviewer's suggestion, we re-phrased the figure caption to make more clear that the blue line is observed data.

It seems that P-E-R misses some of the high-frequency behavior in the gravity residuals. Can you comment? Is it because s is slightly too high for near-surface soil moisture change? I.e., when converting from dg to ds, there is a slightly non-linear effect that would make s, as shown in figure 10, slightly less dynamic.

R27: It is true that the constant sensitivity factor of gravimeter s leads to underestimation of peaks related to near-surface soil moisture changes. The mean factor s equals 0.478 nm s-2 mm-1, while s equals 0.441 nm s-2 mm-1 for layer between 0 and 1 cm. Thus, the sudden water storage change can be underestimated by 8 % as long as the water is concentrated on or close to the surface. Furthermore, the v,u,a,c parameters were optimized for whole time series, leading to parameter values that reflect primarily the dominant, i.e., seasonal, variations. Variations at higher frequencies caused by rainfall or discharge may thus be underestimated by these parameter values. This is similar for the evapotranspiration rates, and also was the reason for the separate analysis of daily evapotranspiration factors. This discussion is added to the revised manuscript.

p.19, line 19: Worth noting there are many factors (including all those mentioned), with

the same 24-hour period as ET.

R28: Besides the hydrological diurnal effects, we mention temperature effects, Earth tides and ocean loading as other factors. Atmospheric tides (S1/S2/...Sn) are also relevant at this period. We stress the problem of overlapping processes with daily period for resolving ET in the revised manuscript.

line 23: State explicitly why day-to-day ET can't be determined directly from the data (i.e., rearrange eq. 3 to solve for (Eref – c) without stacking or a moving window. You mention the main obstacles earlier, but it's not obvious these are the exact (and only) reasons it can't be done. line 30 would indicate the estimate improves as the window gets even shorter than 9 days?

R29: The main reason for solving the equation within a moving window is to reduce the influence of gravimeter noise and non-hydrological effects on the estimated parameter values. This issue is mentioned in the manuscript. Gravimeters do not measure the rate of change per day, but the signal is rather equivalent to a cumulative sum. To solve directly for daily ET rates as measured by the lysimeter, differences between days need to be considered. However, the differences are highly sensitive to any noise of instrumental or geophysical origin. Feeding the moving window with an a-priori estimate of potential/reference ET acts like a physical low-pass filter that minimizes the noise effects. The longer the window, the higher the reduction of noise. On the other hand, however, the shorter the window length the higher the accuracy at daily level. This holds true, however, for the ideal situation when comparing potential and actual ET time series without other sources of variations. The reduction of the window to 1 day would lead to an errorless estimation. Thus, a value of 11 was chosen for the window length as a trade-off between noise mitigation and accuracy. This explanation is added to the revised manuscript.

p. 20, line 17: sign not sing

R30: Corrected.

I find the discussion about the E discrepancy very interesting, as it's not obvious to me how a gravimeter at the land surface responds to ET – to decrease gravity seems to require advecting the evaporated water away from the region of sensitivity.

R31: Indeed, the analysis and discussion so far assumes that the evaporated water is transported away from the region of influence of the gravimeter (or that its effect can be mapped by the atmospheric pressure data). Another reason for discrepancies between gravity-based ET and lysimeter-based ET might be that this assumption does not hold due to unresolved atmospheric transport processes. We mention this point in the discussion.

p.22, line 9: I'm unclear what a "dedicated sensitivity" is. Is it appropriate to cite a paper in preparation?

R32: We remove the term 'dedicated'. The idea of the sentence remains unchanged. At the time of writing the manuscript, we included the study in preparation while expecting it to be further advanced at the time of publication of the present manuscript. We may need to remove Reich et al. from the final version of this paper if not yet published.

line 21: I'm not sure "fundament" is widely known (it's unfamiliar to me). Also, the significant limitation of requiring AC power.

R33: "Fundament" replaced by "monument". Power requirements added as a further limiting factor.